RESEARCH CULTURE

# A survey of early-career researchers in Australia

**Abstract** Early-career researchers (ECRs) make up a large portion of the academic workforce and their experiences often reflect the wider culture of the research system. Here we surveyed 658 ECRs working in Australia to better understand the needs and challenges faced by this community. Although most respondents indicated a 'love of science', many also expressed an intention to leave their research position. The responses highlight how job insecurity, workplace culture, mentorship and 'questionable research practices' are impacting the job satisfaction of ECRs and potentially compromising science in Australia. We also make recommendations for addressing some of these concerns.

**KATHERINE CHRISTIAN\*, CAROLYN JOHNSTONE, JO-ANN LARKINS, WENDY WRIGHT AND MICHAEL R DORAN\***

**\*For correspondence:**
katherinechristian@students.
federation.edu.au (KC); michael.
doran@qut.edu.au (MRD)

**Competing interests:** The authors declare that no competing interests exist.

## Introduction

Advances in science, technology, engineering, mathematics and medicine (STEMM) have revolutionized virtually every facet of modern life. In Australia the government is relying on further advances in these fields to underpin future economic prosperity (*Innovation and Australian Government, 2017*). Australia has also become the largest provider of education to international students in the Organization for Economic Co-operation and Development (OECD) nations (*Sá and Sabzalieva, 2018*), with a rise in the number of PhD students accounting for a large portion of this increase: indeed, the number of students (domestic and international) completing a PhD in 2019 was more than twice the number for 2000 (*McCarthy and Wienk, 2019*).

Two international surveys conducted in 2015 (*Ghaffarzadegan et al., 2015*) and 2017 (*Woolston, 2017*) indicated that nearly 78% and 75% of PhD candidates, respectively, aspired to obtain a job in academia, despite the global lack of such job opportunities. Not all PhD graduates need work in academia, but the advanced industries that typically employ highly skilled workers are less developed in Australia than, say, the United States or Germany (*Christopherson et al., 2014*; *Weller and O'Neill, 2014*). Australian graduates are therefore more dependent on academia as an employer than graduates from other OECD nations. A previous survey of 284 postdoctoral researchers in Australia revealed that more than half (52%) took their position hoping to transition to a full-time research role in academia (*Hardy et al., 2016*). The majority of respondents (54%) felt that structural, rather than personal limitations would prevent them from realizing a long-term research career. In addition to concerns about the international so-called 'glut' of PhD students (*Woolston, 2017*; *Woolston, 2014*; *Woolston, 2019*) there have been concerns about the reproducibility of published findings in a number of research areas (*Baker, 2016*; *Begley and Ellis, 2012*).

Early-career researchers (ECRs) represent the transition stage between PhD and senior academic positions, and their well-being provides insight into the health of the industry. In this study, we surveyed ECRs in STEMM disciplines in Australia to better understand the pressures impacting them and their career development. We defined ECRs as being less than 10 years since PhD completion, similar to the definition used by the Global Young Academy in their study of how to best support young scientists on a global scale, and another important survey of the STEMM workforce conducted in Australia

(*Pain, 2014*; *Bell and Yates, 2015*). Data were collected from respondents employed in research institutions or universities via an on-line survey (n = 658), which was developed based on previously published questions and through focus group discussions (*Supplementary file 1*). From our survey we quantified job satisfaction, likelihood of continuing to work in research in Australia, views on mentoring and career planning, and observation of questionable research practices (see material and methods for more detail on how topics were selected for the survey).

In addition to fraud, John et al., popularised the notion that *questionable research practices* included less egregious practices, such as data exclusion, may in fact be more prevalent and more damaging to the academic enterprise (*John et al., 2012*). Their findings warned that the frequency of questionable practices may be so prolific that they are becoming 'the norm' in research. Note that we did not define 'questionable research practices' in our survey. However, our data suggests that the systemic pressures compromising the training and career progression of ECRs in Australia may also contribute to a decline in research quality. It is time to carefully consider if the support and career advancement options available to ECRs in STEMM subjects is aligned with Australia's scientific aspirations. As many of the documented pressures highlighted in this study are common global problems, these data likely highlight important considerations relevant to the international research community.

## Results

### Demographic of respondents

Of the 658 respondents, 65.8% identified as female and 34.2% as male. The two most common age brackets were 31–35 years old (42.7%) and 36–40 years old (25.9%), with most respondents having completed their PhD 2–4 years earlier (37.8%) or 5–7 years earlier (25.3%). The four most common countries of birth were Australia (50.6%), England (6.2%), India (4%) and China (2.5%). Of the respondents, 48% identified as being in the medical and health sciences and most (63.2%) were employed in a research only position. Recent data from the Australian Research Council (ARC) indicates that 38.9% of Australia's STEMM workforce is employed in the medical and health sciences (*Table 1*; *Australian Research Council, 2019*). Comparison of our survey demographics with this ARC data indicates that our sample and the target population were not statistically different by discipline (chi square = 16.344, df = 9, p=0.06), and our survey population can be considered representative. A more detailed summary of respondent demographics is provided in *Figure 1*.

**Table 1.** Distribution of research disciplines in STEMM.
The percentage of academics in Australia that work in different STEMM disciplines, relative to the percentage of survey respondents in each discipline (n = 658). \*\*Australian work force data sourced from *Australian Research Council, 2019*.

| Discipline | \*\*Percentage of Australian academic STEMM workforce | Percentage of respondents to this survey |
|---|---|---|
| Mathematical Sciences | 3.8% | 2.8% |
| Physical Sciences | 4.3% | 8.1% |
| Chemical Sciences | 4.7% | 5.7% |
| Earth Sciences | 3.5% | 3.0% |
| Environmental Sciences | 3.2% | 4.0% |
| Biological Sciences | 12.6% | 20.9% |
| Agricultural and Veterinary Sciences | 4.5% | 1.4% |
| Information and Computing Sciences | 6.9% | 2.2% |
| Engineering | 15.4% | 3.6% |
| Technology | 2.1% | 0.8% |
| Medical and Health Sciences | 38.9% | 47.5% |

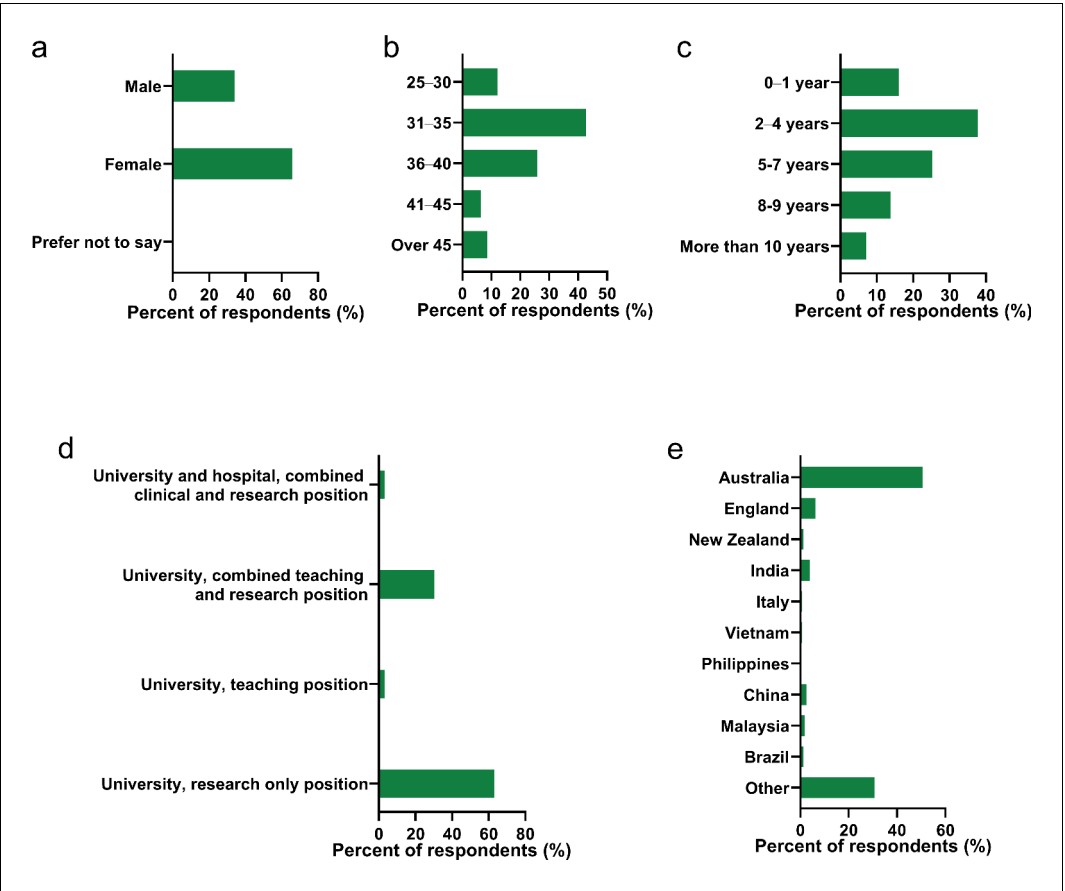

**Figure 1.** Demographic data of survey respondents. (**a**) Gender of respondent n = 658. (**b**) Age of respondent n = 660. (**c**) Years since completion of PhD n = 833 (ineligible respondents subsequently terminated). (**d**) Nature of employment n = 638 (does not include 'other'). (**e**) Country of birth n = 658.

### Characteristics that influence ECR job satisfaction

We attempted to identify workplace characteristics that influenced ECR job satisfaction and career progression. Using a non-biased approach, we used text responses to generate a word cloud (*Figure 2*), and tabulated the most common words associated with satisfaction or dissatisfaction in the context of the responses to survey question 76 (*Why do you stay?*). Respondents almost universally noted their '*love*' of research and the job fulfilment it provides (*Figure 2—source data 1* and *Table 2*). ECRs reported that they derived fulfilment from research, mentoring, teaching and the general sense that they are making a meaningful contribution to society, while job security and challenges associated with the job remain major concerns (*Table 2*). One respondent said, 'I love it! I am passionate about my work and driven to make a difference. I will keep going as long as I can'.

### Satisfaction with workplace culture and intention to leave academia

We queried ECRs regarding satisfaction with their workplace culture. Academic workplace culture, which encompasses interactions between colleagues and professional norms (*Faulkner, 2009*), has evolved with corporate pursuits and hypercompetitive funding environments (*Edwards and Roy, 2017*). *Figure 3* shows that 51.0% of respondents indicated that they were satisfied or very satisfied with their workplace culture, while a concerning 31.9% were somewhat or very dissatisfied with their workplace culture. Overall, the survey data indicated that the most significant barrier to job satisfaction and career advancement was job insecurity (48.9%). A poor workplace culture (31.9% dissatisfied or very dissatisfied), lack of support from institutional superiors (60.1% a problem or significant problem), poor leadership and management (33.1% dissatisfied or very dissatisfied), and lack of recognition (22.6%

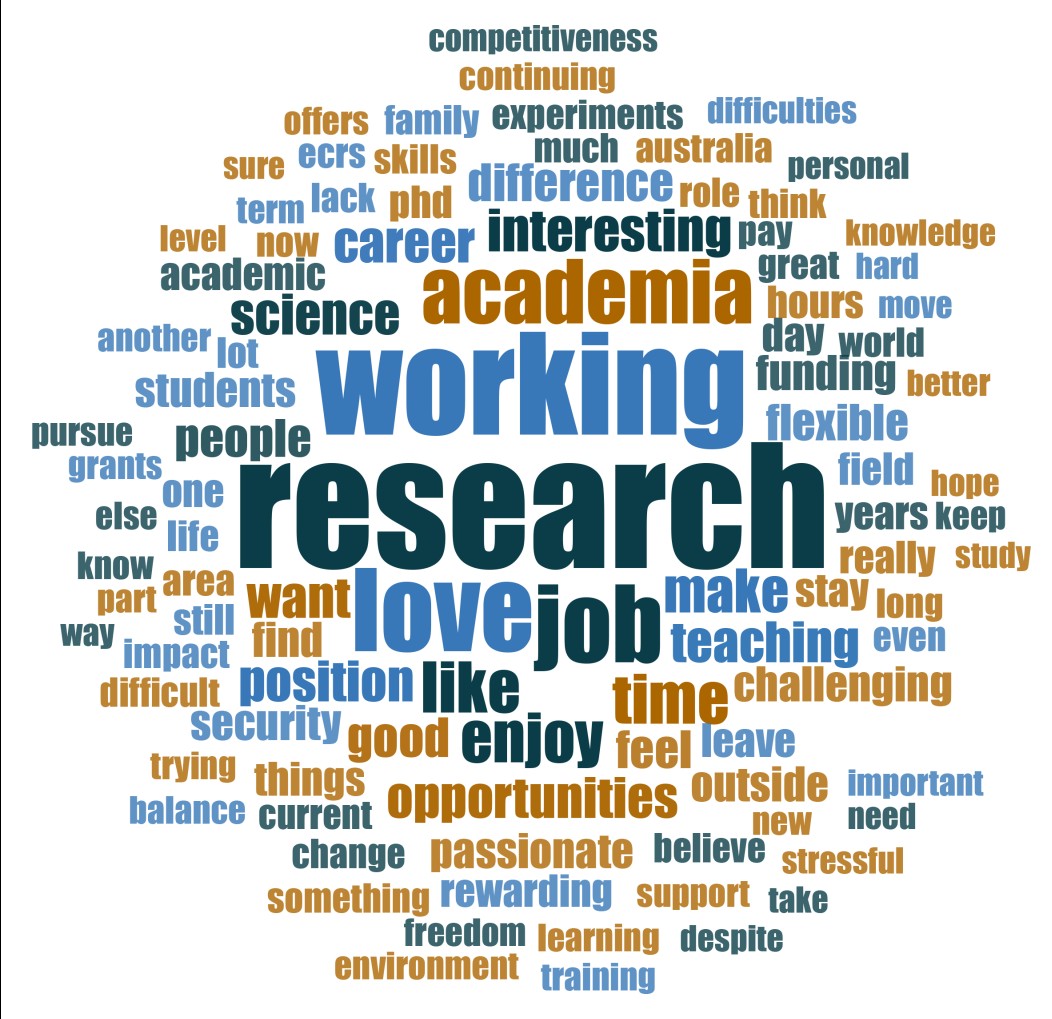

**Figure 2.** Why respondents stay in research. Word cloud of the responses to survey question number 76 (*Why do they stay in science?*). The analysis tool NVivo v12 for Mac was used to count the frequency of words in the answers. Of the 334 answers, 108 mentioned love, 16 mentioned passionate and 11 mentioned passion (see *Figure 2—source data 1*).

The online version of this article includes the following source data for figure 2:

**Source data 1.** These data were generated from the open ended responsed to question 76: Why do you choose to stay in science?

disagree or strongly disagree they feel valued) and lack of work life balance (38.1% disagree or strongly disagree they can manage the demands of work and home life) were further major influences compromising satisfaction. When respondents positively described job satisfaction, this was associated with responses indicating good leadership and management (47.3% were satisfied or very satisfied), feeling valued (61.9% agree or strongly agree they feel valued) and confidence in job prospects (35% agree or strongly agree they have good job prospects).

Some respondents indicated that attitudes to their gender (13.1% reported a problem or major problem), ethnic background (5.8%) or sexual orientation (1.4%) may have impacted their career advancement. However, when we made detailed queries regarding workplace challenges detailed in Question 36, we observed effectively no significant difference in the responses from male or female researchers, nor from researchers who were Australian born or non-Australian born. We were unable to draw conclusions regarding the impact sexual orientation might have on workplace challenges, as we did not ask respondents to identify their sexual orientation other than with respect to their living arrangements.

**Table 2.** Selected responses to the question: Why do you choose to stay in science? (question 76 in our survey).
Quotes were selected as they conveyed respondents' love of science. In addition to the positive responses shown here, respondents also expressed concerns about job security, mentorship and workplace culture.

| Quote number | Specific response |
|---|---|
| 1 | *I love figuring stuff out. I love inventing new ways to measure stuff.* |
| 2 | *I love it! I am passionate about my work and driven to make a difference. I will keep going as long as I can.* |
| 3 | *I love my job - it doesn't feel like a job - I get to do what I enjoy. That said, the lack of job security and the challenges of having a family, buying a house and staying in the one city in Australia makes it difficult to imagine remaining in research/academia.* |
| 4 | *I love my job, being able to develop new research questions and work with clinicians and patients. But I do not love the industry. The lack of job security, challenges in supporting a team, and constant pressure to do more as soon as you can is deeply problematic.* |
| 5 | *I love research and discovery, a core part of my identity is 'scientist'. I'm not sure who I would be outside academia.* |
| 6 | *I love research and I love teaching, and academia offers the opportunity for both of these. Improved job security would be the one key thing to improve my experience.* |
| 7 | *I love research and my research area, I want to help people through my science discoveries and the sharing of these results.* |
| 8 | *I love research! No two days are the same and it is extremely rewarding. You have to celebrate the few good days you have (manuscript accepted, award at a conference, grant etc.). The opportunity to truly make a difference to the lives of people is what keeps me going!* |

Those who were more than 4 years post-PhD were less likely to be satisfied with their job (55.7%) compared to those who were 4 years or less post-PhD (66.9%). Similarly, those who were more than 4 years post-PhD tended to indicate a higher frequency of being negatively impacted by lack of support from institutional supervisors (increase of 13.4%), questionable research practices of colleagues within their institution (increase of 14.5%), and harassment based on power position (increase of 5.6%). In addition, less than 40% express satisfaction with leadership and management in their workplace compared to 53.4% of respondents who are less than 4 years post-PhD. In responses to Question 74–1 'This is a poor time for any young person to begin an academic career in my field', more senior postdoctoral researchers indicated that this was not a good time to be in science, and were less willing to recommend science as a career (73.2% compared to 58.5% of junior researchers).

We compared our survey respondents' satisfaction data with previous survey data from academics in Australia (*Supplementary file 2*; *Bell and Yates, 2015*; *Coates et al., 2009*; *Bexley et al., 2011*; *NTEU State of the Uni Survey, 2017*). Each of these studies used one or more of the 'job satisfaction' questions from our survey in their own survey of the academic workforce in Australia. It can be seen respondents from the current study are more concerned about job security than respondents in any of the other studies. Our respondents also indicated a higher level of personal stress (52%) than those in all the other studies (28%–43%)

and agree most strongly (65%) 'this is not a good time for any young person to aspire to an academic career'. Their reported job satisfaction is low (62%). The combination of answers to these questions for current ECRs relative to those for the other studies indicates a situation about which there should be grave concern.

Previous studies have identified diversity and inclusion as factors that have impact on senior academics' dissatisfaction (*Zimmerman et al., 2016*; *Professionals Australia, 2014*), including the career progression for female academics (*Potvin et al., 2018*; *Else, 2019*; *Gewin, 2018*). However, in our survey of ECRs working in Australia, most identified as satisfied or at least unconcerned, regarding discrimination with respect to age (87.2% satisfied or not concerned), gender (85.9% satisfied or not concerned), ethnic background (93.8% satisfied or not concerned) or sexual orientation (98.2% satisfied or not concerned). Low levels of concern regarding attitude to ethnicity could reflect an under-representation of respondents from minority backgrounds. Similarly, low rates of reported concern about attitudes to sexual orientation may reflect the small number of respondents (3%) who identified as living with a same sex partner.

When asked to what extent they agreed with the statement 'I am satisfied with my workplace's commitment to a diverse and inclusive workplace', 6.4% strongly disagreed, 11.5% disagreed, 20% neither agreed or disagreed, 41.2% agreed and 21% strongly agreed. Gender did not appear to influence ECR's perception of their workplace, with satisfaction rates being

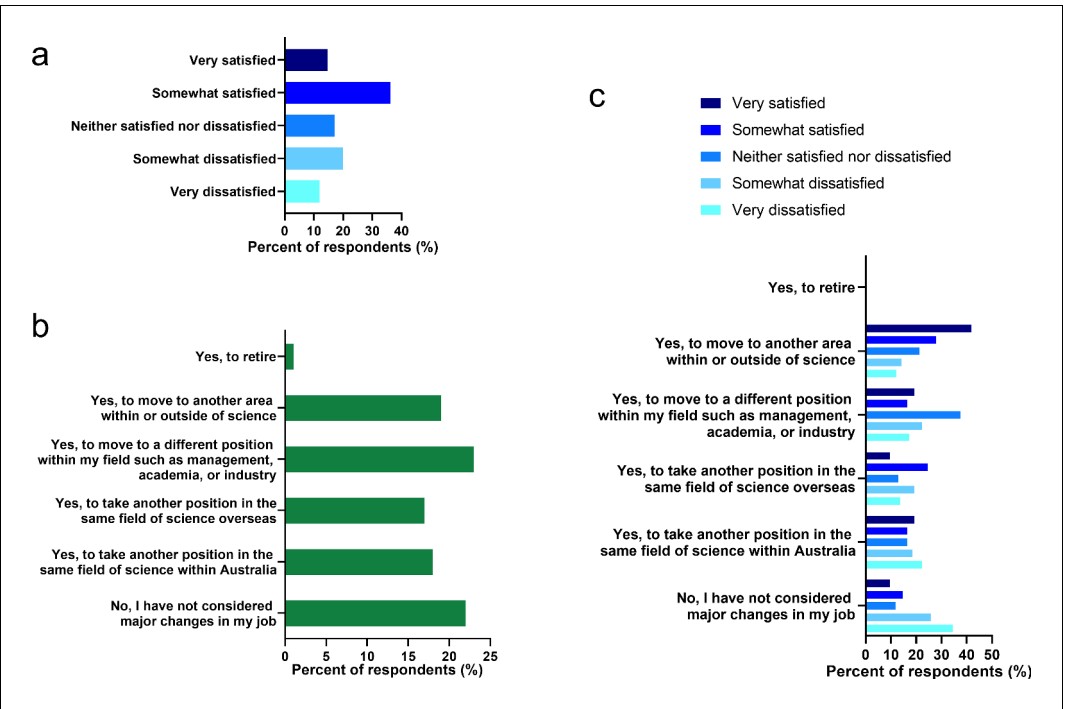

**Figure 3.** Job satisfaction does not influence the decision to make a major career change. (a) Respondents were asked to rate their overall satisfaction with their current work (Question 31–4 in survey, n = 566). (b) Respondents were asked if within the last five years they had considered any major career or position changes, and what these might be (Question 61 in survey, n = 470). (c) For those considering a major career or position change in the previous 5 years, we stratified responses from respondents based on satisfaction with their current position (n = 470).

The online version of this article includes the following figure supplement(s) for figure 3:

**Figure supplement 1.** Satisfaction with workplace culture stratified by gender and country of birth.

similar between male and female respondents (50.3% satisfied or very satisfied versus 51.9%, *Figure 3—figure supplement 1A*). However, workplace position did influence satisfaction rates, with those on teaching only positions reporting the highest levels of satisfaction (23.1%), and none reporting being very dissatisfied (*Figure 3—figure supplement 1B*). Those in research and teaching positions reported high levels of dissatisfaction (25.4% somewhat dissatisfied and 12.7% very dissatisfied).

We did not identify an obvious influence on country of birth on job satisfaction (*Table 3* and *Figure 3—figure supplement 1C*). Both Australian and non-Australian born researchers reported finding their work rewarding (78.0% and 76.4%), were satisfied or very satisfied with their current job (65.2% and 59.5%), but also categorised their job as a considerable source of personal strain (56.2% and 45.9%) and indicated a lack of support from institutional supervisors (63.5% and 55.6%). We regard the consistency in reporting as not necessarily indicating that

country of birth has no impact on job satisfaction, but rather that systemic workplace culture issues dominate the concerns of both Australian and non-Australian born researchers.

We asked if ECRs had considered a major career or position change in the previous 5 years. The majority (78.3%) of respondents had considered a major career change, while only 21.7% had not (*Figure 3B*). Many considered leaving academia all together (19.1%) or moving overseas (17.4%) in order to progress their career path. For each group of respondents that indicated that they had considered a major career change, we quantified how satisfied they were with their current work environment (*Figure 3C*). Interestingly, within the population of ECRs who had not considered a career change, the largest group (34.5%) were dissatisfied with their current workplace. By contrast, within the population of ECRs who indicated that they had recently considered moving to another area within or outside of science, the largest group (41.9%) were very satisfied with

**Table 3.** How does country of origin influence job satisfaction?
Table shows the percentage of respondents born in Australia and born outside Australia who agreed with the following statements (under Question Detail) about their job satisfaction.

| Question detail | Australian born | N | Not born in Australia | N | All |
|---|---|---|---|---|---|
| I am satisfied with the attitude to people of my ethnicity | 48.7% | 263 | 44.2% | 249 | 46.4% |
| Overall, I find my work rewarding | 78.0% | 287 | 76.4% | 271 | 77.2% |
| I am satisfied with the culture of my workplace | 53.0% | 287 | 49.3% | 270 | 51.0% |
| I have been impacted by harassment based on power position | 32.7% | 263 | 34.4% | 249 | 33.5% |
| I have been impacted by lack of support from institutional supervisors | 63.5% | 263 | 55.6% | 247 | 59.8% |
| I have been impacted by questionable research practices of colleagues within my institution | 36.1% | 263 | 39.7% | 247 | 37.1% |
| I am satisfied with the leadership and management of my workplace | 48.8% | 287 | 45.9% | 270 | 47.1% |
| My job is a source of considerable personal strain | 56.2% | 242 | 45.9% | 220 | 51.6% |
| How would you rate your overall satisfaction with your current job (satisfied or very satisfied) | 65.2% | 242 | 59.5% | 220 | 62.3% |

their current workplace. These data suggest that there might be populations of ECRs who are unhappy in their current workplace, but feel trapped, while there is another population of ECRs who are very happy in their current workplace, but feel changing jobs would be beneficial. More generally, ECR's satisfaction with their current position does not appear to significantly bias their consideration of major career changes.

### Influence of gender, country of origin and appointment type on workplace challenges

Workplace and career progression challenges are displayed in *Table 4*. Data were sorted based on gender and subsequently sorted based on appointment types, which were categorized as '*research only*', '*research and teaching*', or '*clinician researcher*'. Those with a teaching or clinical appointment are likely to be less dependent on research funds for their salary, and thus their perspectives may differ. Greater than 50% of both male (52.4%) and female (63.8%) ECRs indicated that they felt they had been negatively impacted by a lack of support from institutional leaders (*Table 4A*). Female ECRs indicated higher rates of inequitable hiring practices (40.0% females versus 35.4% males) and harassment from those in a position of power (31.7% females versus 25.9% males). Interviews with ECRs conducted in another part of this project and the focus group which evaluated the questionnaire for this survey, as well as survey responses, suggest instances where senior academics (both male and female) were regarded as bullies (the results of this part of the project will appear in a separate publication). When asked if they feel safe in the work

environment, overall 12.5% felt unsafe with an unexpected bias of males (15.6%) to females (11.0%) reporting this problem (*Table 4A*). We further delineated these data based on researchers who were either Australian or non-Australian born, finding that non-Australian born researchers reported being marginally less safe at work (15.1%) than respondents born in Australia (10.1%). Similarly, slightly more non-Australian born researchers reported inequitable hiring practices being a significant problem in their career advancement (14.5% versus 11.8%).

### Impact of inappropriate behaviours

Particularly concerning was the number of female and male ECRs who identified that their wellbeing, or their career had been impacted by *questionable research practices* within their institution (41.4% of females and 30.7% of males) or external to their institution (33.6% of females and 28.9% of males). While some respondents would have been cautious not to reveal specifics regarding questionable research practices, even in a confidential survey, a number of comments did provide reasonably detailed examples of concerning behaviour (*Table 5*): '…what they wanted to see result-wise wasn't what I was seeing. And so, I was being accused of misconduct because I wasn't seeing what they wanted me to see, and I wouldn't change that'.

When the data was re-sorted based on appointment type, it was possible to estimate the influence that different appointments and contract stability may have on ECR job satisfaction and/or career progression (*Table 4B*). The majority of clinician researchers (79.0%) reported having been impacted by lack of support from

**Table 4.** How gender and academic position affect job satisfaction and career advancement.
(A) Factors that impacted ECR job satisfaction and/or career progression, analysed with respect to gender (n = 511). (B) Factors that impacted on ECR job satisfaction and/or career progression, analysed with respect to ECR appointment type (n = 509). Teaching only (20) and 'Other' (62) responses are omitted from (B).

**(A)**

| Workplace characteristic | Female (n = 345) | | | | Male (n = 166) | | |
|---|---|---|---|---|---|---|---|
| | Impacted | Strongly impacted | Total impacted | | Impacted | Strongly impacted | Total |
| Lack of support from institutional superiors | 45.5% | 18.3% | 63.8% | | 34.3% | 18.1% | 52.4% |
| Inequitable hiring practices | 27.8% | 12.2% | 40.0% | | 19.8% | 15.6% | 35.4% |
| Harassment based on power position | 25.4% | 11.6% | 37.1% | | 14.5% | 11.4% | 25.9% |
| Questionable research practices of colleagues within their institution | 34.2% | 7.2% | 41.4% | | 18.7% | 12.0% | 30.7% |
| Questionable research practices outside their institution | 27.2% | 6.4% | 33.6% | | 21.7% | 7.2% | 28.9% |
| Feeling unsafe in the work environment | 4.3% | 6.7% | 11.0% | | 7.0% | 8.6% | 15.6% |

**(B)**

| Workplace characteristic | Research only (n = 282) | | | Research and teaching (n = 126) | | | Clinician researcher (n = 19) | | |
|---|---|---|---|---|---|---|---|---|---|
| | Impacted | Strongly impacted | Total | Impacted | Strongly impacted | Total | Impacted | Strongly impacted | Total |
| Lack of support from institutional superiors | 37.4% | 17.1% | 54.5% | 42.1% | 22.2% | 64.3% | 63.2% | 15.8% | 79.0% |
| Inequitable hiring practices | 23.8% | 9.6% | 33.4% | 26.2% | 20.6% | 46.8% | 42.1% | 10.5% | 52.6% |
| Harassment based on power position | 20.3% | 11.4% | 31.7% | 27.0% | 14.3% | 41.3% | 15.8% | 10.5% | 26.3% |
| Questionable research practices of colleagues within their institution | 27.0% | 10.3% | 37.3% | 27.8% | 6.3% | 34.1% | 26.3% | 10.5% | 36.8% |
| Questionable research practices outside their institution | 25.3% | 8.5% | 33.8% | 23.8% | 4.0% | 27.8% | 21.1% | 15.8% | 36.9% |
| Feeling unsafe in the work environment | 8.4% | 4.2% | 12.6% | 6.7% | 7.4% | 14.1% | 5.0% | 5.0% | 10.0% |

institutional superiors, compared with research and teaching ECRs (64.3%), and research only ECRs (54.5%). This pattern was replicated with respect to inequitable hiring practices reported more frequently by clinician researchers (52.6%), followed by research and teaching ECRs (46.8%), and research only ECRs (33.4%). These data may indicate that ECRs employed across multiple research, teaching and clinical departments struggle more to find unified institutional support, or to access what they perceive to be equitable hiring/recruitment practices. While clinician researchers we surveyed faced a number of challenges, we note that our survey only captured data from 19 such respondents. These ECRs, in many cases, rely primarily on their clinical appointment as a source of income, and so are potentially less sensitive to job insecurities felt by research only ECRs. Only two (10%) of clinician researcher ECRs reported feeling unsafe at work, compared with 39 (12.6%) research only ECRs, and 19 (14.1%) research and teaching ECRs. Similarly, clinician researchers reported

less impact of harassment based on power positions (26.3%), compared to research only (31.7%) and research and teaching ECRs (41.3%). It is possible that the job security benefits realised by clinician researchers manifests itself in actual or perceived reductions in feeling unsafe at work, and reduced harassment from those in a position of power.

The frequency that *questionable research practices* had negatively impacted ECRs declined incrementally from those who were research-only (37.3% internally and 33.8% externally), clinician researchers (36.8% internally and 36.9% externally) and research and teaching (34.1% internally and 27.8% externally). These data suggest that greater research time commitment may increase the frequency of exposure to *questionable research practices*, but that the stability associated with salary funding from a teaching or clinical position does not obscure the perception that this is a major problem.

**Table 5.** Quotes regarding *questionable research practices* (from surveys and interviews).

| Quote number | Specific response |
|---|---|
| 1 | ….the bullying and stuff came to a head and the scientific work was looked at because this person had brought up kind of bullying and harassment allegations against the supervisor. So they in turn looked at the work that this person had been doing and they'd been falsifying… |
| 2 | Lack of funding and the need to 'sell' your research, often leads to many researchers fabricating and embellishing data. This leads to the inability of genuine researchers to replicate findings, wasting precious time and resources, giving up and then their contracts not being renewed because the boss doesn't get the 10 publications per year they demand. |
| 3 | I believe that the whole Academia environment is corrupted and has lost its true vision. The lack of funding is making researchers to sometimes make-up data to get grants or to publish meaningless papers just for the sake of raising the numbers. |
| 4 | being used by post docs and high level senior researchers' who take credit for your research work ideas and use info in your recruitment applications unethically for themselves…bias recruitment towards international students and overseas post docs who are extremely competitive and who want to get permanent residency and who also bully harass local students and researchers' to take over their research and jobs. |
| 5 | …what they wanted to see result-wise wasn't what I was seeing. And so I was being accused of misconduct because I wasn't seeing what they wanted me to see, and I wouldn't change that. |
| 6 | Not saying, 'do this' but pressure to – if something were to fail to almost keep saying, 'Do it again, do it again, do it again, do it again'' in order to get you to make it work. And those people have just said, 'No, it doesn't and I'll spend the whole year repeating it but it's not going to change the outcome'. |
| 7 | Q But are they getting their names on because they've actually been involved? Are we flouting the convention here?<br>A They haven't done anything.<br>Q So his investment in them is…<br>A Is purely so they can get grant funding through having papers. |

### The need to relocate

Many academics relocate to capture job opportunities, and many appreciate the opportunity to move internationally with their career. However, we observed that the academic culture promotes a perceived *need* to relocate during the ECR years, and that many ECRs who wished to remain in academia considered moving as part of their career development process. To better understand this phenomenon, we asked more detailed questions regarding decisions to move. The answers to these questions indicated that moves to new institutions can be stressful, are frequently made without financial compensation, and can be challenging for families and for careers (*Table 6*).

This problem was highlighted in recent article published in Science, which described the struggles of a tenure-track academic on a work visa in the United States who was unable to gain financial approval to purchase a home (*Evaristo, 2020*). While a tenure-track academic can make long-term decisions, this is virtually impossible for many ECRs. Most (68.1%) respondents reported that they had already changed location in order to advance their careers. Of these, 28.6% of ECRs had moved once, 20.1% had moved twice and 19.5% had moved more than twice. Commonly expressed consequences, noted in interviews and in text-based responses, were that relocation was associated with stress, separations from family, loss of support network, personal cost and loss of career momentum. Within the small number of respondents that reported a chronic health condition (12%) some indicated that relocation was challenging.

### Mentorship and career guidance

To better understand ECRs concerns regarding support from institutional leaders, respondents were asked to describe their mentorship and career guidance. A definition of a of mentor was provided with the questions: "A mentor is someone who is there to assist you achieve your personal, academic and career exploration goals. This person is not necessarily your supervisor'. In our survey, 61.9% of ECRs reported having a mentor, while 38.1% did not. We asked ECRs to indicate what aspects of mentoring they valued most, and these data are summarized in *Figure 4A*. ECRs valued advice on career decisions (81.7%) as the most important contribution from mentors. This was followed by integration into networks (77.2%), and direct influence on their gaining employment (56.7%). Ranked less significant, but still important, were skill training on methodologies (60.3%), fundraising (50.8%), and scientific writing (59.7%). Of those with a mentor, the quality of the mentoring was often described as inadequate, and some indicated that they paid for external mentoring. From the survey data (n = 322), those who did receive

**Table 6.** Quotes regarding the stress of relocation.

| Quote number | Specific response |
| --- | --- |
| 1 | *The most significant impact has been on my productivity for the few months after I move. Settling into a new environment takes time. I had little to no support to find accommodation[sic], so much of my time was spent on this. The mental/emotional drain of a move is also significant.* |
| 2 | *Starting from scratch with a whole new group of colleagues who don't know you and struggling to find research momentum in a new institute, city and country, all of which is very different to previous places you've lived before. Everything is done differently and you're constantly learning the hard way, which takes time and significantly eats into your research progress. It's also lonely and can inhibit the development of long-lasting professional and personal relationships because you have no idea how long you'll really be in the country.* |
| 3 | *Lack of stability, no ability to build long term friendships and networks, relationship breakdowns, financial costs, inability to buy a house.* |
| 4 | *Loss of traction and momentum in science. Loss of family and friend support. Starting life from scratch. Financial loss from moving costs, to higher rents in locations I moved to.* |
| 5 | *Relocation meant my partner having to give up her job* |
| 6 | *Separation from family and friends, impact on spouse's career, new start at new institutions take time and are somewhat unproductive.* |
| 7 | *Moving internationally with a young family has been extremely difficult. Lack of family support with both myself and husband working full time is extremely difficult to manage.* |
| 8 | *Moving to further career progression - like an international fellowship visit - should not be applicable to all fields of research. Furthermore in families with two working adults this is unrealistic and archaic. There are other options to building an international reputation. I moved internationally to complete my PhD.* |

mentoring (Question 44 of our survey) described it as follows; 15.1% neutral, 7.5% not beneficial, 32.8% highly beneficial, or 44.6% beneficial.

With respect to supervision, as opposed to mentoring, only 68.3% of respondents had a performance review in the past two years, indicating that 31.7% had not. While half of the 31.7% respondents with no performance review indicated that they had recently been appointed or were on probation (not unusual in an environment where short term contracts are commonplace), the other half had not been offered a review. Many who did have a performance review did not find the process useful (41.6%; *Figure 4B*). There was no opportunity given to provide an explanation for these answers, however, respondents identified the primary utility of performance reviews as being (1) a review of personal progress (57.1%), (2) identifying strengths and achievements (50.7%), (3) help focusing on career aspirations (50.4%), and (4) to highlight issues (44.2%). ECRs identified performance reviews as least useful in leading to changes in their work practices. Given that performance reviews are often used to influence work practices, it is useful to know that this process is frequently viewed as ineffective.

### Intention to leave
Finally, we circled back and considered if the positions ECRs held were similar to what they had anticipated, and if they intended to remain in or leave these positions (*Figure 5A*). Relatively few (14.5%) found their current position to

better or much better than expected. Regardless of their perception of the position, many ECRs indicated their intention to leave. There was a trend (regression analysis, p=0.0234) indicating a greater bias to leave the position depending on how it had met expectations (*Figure 5B*). However, even in instances where the current position was much better than expected, nearly 40% more (61.5%) ECRs intended to leave the position rather than remain (38.5%).

As most ECR positions are short-term contracts, including those supported by 'soft money' (where all expenses for that researcher, including salary, are covered by fixed-term grants), it might be rational to expect to have to leave a position even if the position had met or exceeded expectations. If ECRs were to leave their current academic position, we asked what the primary motivation would be (*Figure 5C*). Cumulatively, two of the possible responses, lack of funding (28.2%) and job insecurity (48.9%), accounted for 77% of likely motivations for ECRs leaving their current position. Establishing an independent research group is the goal of many ECRs. Lack of independent positions was cited as the motivation 11.8% of ECRs would use to justify leaving their current position. While in *Table 5* many respondents list poor institutional support as problematic, only 1.4% of respondents cite interpersonal relationships with their supervisor as a potential motivation for leaving their current position. We found that family/carer responsibilities were cited by

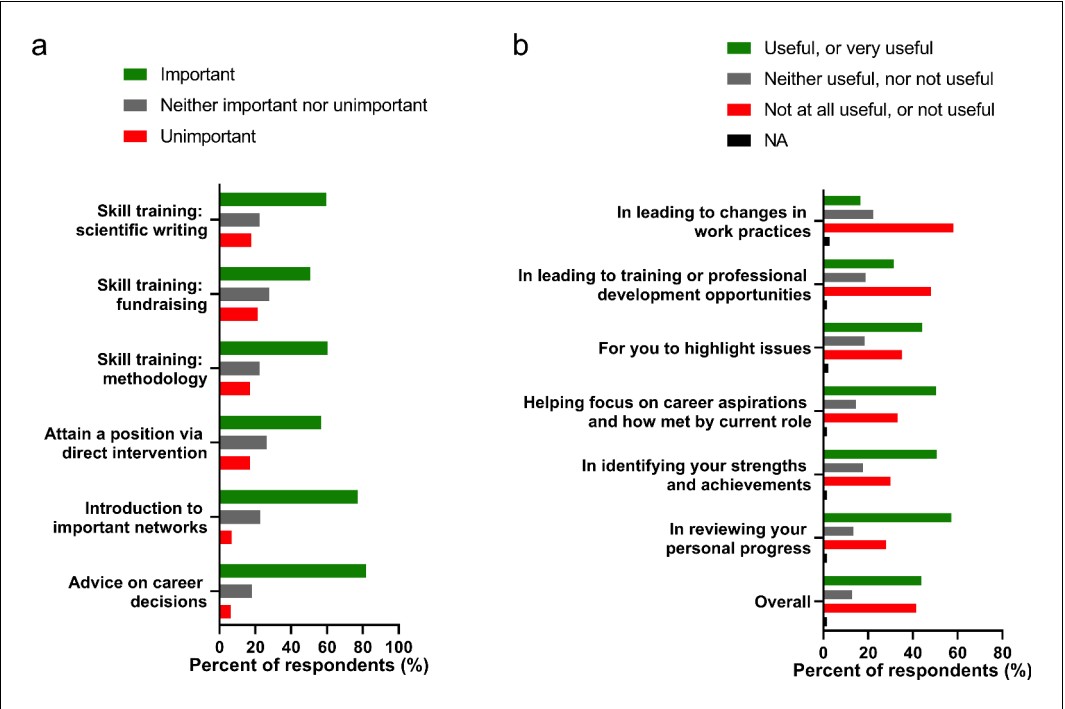

**Figure 4.** Aspects of mentoring that are the most and least important to ECRs. (a) We asked respondents to indicate how much value they placed on different aspects of mentoring from more senior colleagues (n = 481 respondents). (b) We asked respondents who had participated in staff performance reviews to indicate which aspects of the review process they valued (n = 322 respondents who received a review).

9.6% of ECRs as a reason to exit academia. Similar to a previous survey of postdoctoral researchers in Australia, the burden of family/carer responsibilities is heavy on both male and female ECRs, suggesting that young parents (male or female) and their families are not sufficiently accommodated by the current system. In interviews, we did identify young mothers on parental leave struggled to continue to run their laboratories, knowing that their staff depend on them, and continued to write publications while on leave out of fear of falling behind. Quotes in *Table 7* provide insights into stresses felt by ECRs in Australia; we leave the comments to speak for themselves.

Given the many challenges faced by ECRs, their persistence in their endeavours to remain in the academic research workforce is impressive. However, their perceived commitment to academia in Australia may be confounded by limited number of alternative (perceived and actual) employment opportunities outside of academia. A number of comments made by ECRs (*Table 8*), indicate that they consider themselves to be inadequately trained for alternative careers, that there are limited alternatives available, or that they regard leaving academia

as a failure. One respondent said, 'I constantly think about leaving academia/research (from necessity not choice) but don't know how and am not qualified for any other jobs.'

## Discussion

It is common vernacular to say that '*ECRs are the future*'. If this is factually true, then are we content with how we are shaping this future? We suggest that this survey data provides reason to be concerned. While ECRs in our survey overwhelmingly and repeatedly indicated that they '*loved*' their work, only 51.0% of ECRs indicated that they were satisfied with their workplace culture. More than half of ECRs felt they had been negatively impacted by a lack of support from institutional leaders. This is in agreement with previous studies which showed that academics loved their work and realised intellectual satisfaction, but were frequently discontented with their own institution and wonder if they would be happier somewhere else, in a different profession or industry (*Johnsrud and Rosser, 2002*; *Smith, 2020*).

Many ECRs in our survey indicated that they did not have a mentor (38.1%), nor performance review (31.7%). Superficially, these data suggest

**Table 7.** Quotes regarding stresses in the current system (explanations offered for responses to Question 73).

| Quote number | Specific response |
|---|---|
| 1 | *I just find the other aspects of the job and the pressure to perform very difficult. I feel like there is a big clock ticking, and my productivity is always being judged relative to the steady ticking of that clock regardless of the ups and downs and other life circumstances.* |
| 2 | *I just wish that the environment didn't feel so pressured and competitive. I have seen so many great ECRs leave research because of the challenges of finding work, meeting expectation, attracting grants. I think the field is too competitive and does not take care of our ECRs and we are poorer for it.* |
| 3 | *I am currently looking outside academia to get away from the culture of harassment... it takes too much of a toll on my health... but I would stay in academia if I were to find a position that didn't subject me to harassment by a supervisor.* |
| 4 | *Job security is based on churning out a large quantity of publications, regardless of quality.*<br>*Three-year fixed-term contracts are very short. In the first 2 years, I focus on my research, however, in my final year, I am thinking about where I am going next. It takes a lot of time and effort to find something else within the research field. I find having an 'exit strategy' important.* |
| 5 | *Having said that, the pressures of the job have considerably increased in the last ten years and the general expectation is that you should work outside normal working hours, without getting paid extra... And that being able to work in academia is a privilege, so one should do whatever it takes to continue in Academia. In my opinion this is a very distorted and dangerous vision, which puts lots of pressure on ECRs, in particular women who are usually starting families at this stage in their careers.* |
| 6 | *At the point of my career, where I am trying to expand my group to potentially have an independent research group, the stresses around funding are a considerable issue for me (as for everyone else, probably). While I have been relatively successful with funding, I feel the pressure of having to support not only my own research, but also the research of those who work with me, and that holds me back from pursuing opportunities that are available to me as I don't want my group to expand too quickly. It also means that I put up with being paid on a lower pay scale than I should be, rather than going for promotion, because I want to conserve funding. This is certainly a constraint on my ability to expand my career prospects.* |
| 7 | *The personal toll it takes to have an academic position is immense. The job insecurity, being unable to plan for anything beyond 1-maybe 2 years is debilitating. Constantly responding to this opportunity, and that opportunity, doing good clever work and being available at all times is tough beyond measure. Not knowing if all this personal sacrifice and tough hard work are even going to be worth it is downright demoralizing. It might all work out, and it might not - but when do you pull the pin??* |
| 8 | *Mental health of ECRs is overlooked and the universities treat us as second class employees that are disposable.* |

that allocation of a mentor and performance review would lead to considerable improvements. However, a number of respondents (41.6%) indicated that they did not find the performance review useful. When mentoring and reviews were provided, ECRs valued career advice most, followed introduction to important networks, and the capacity of their mentor to directly help them find employment. Ranked less significant, but still important, were skill training on methodologies, fundraising, and scientific writing. These preferences may seem surprising, but a previous survey of postgraduate researchers in Australia found that the quality of supervision did not positively influence initial job attainment, but that '*nurturing networking and careers advice*' did (*Jackson and Michelson, 2015*). This pattern may remain robust among STEMM ECRs in Australia, where '*who you know*' could play a significant role in employment outcomes. Our data suggest that ECRs believe this is a factor, and many report being impacted by inequitable hiring practices (40.0%, females and 35.4%, males). Job stress in the sector is likely causing similar patterns to evolve in jurisdictions around the world (see discussion on social networks and so call 'gate keepers') and

academic recruitment (*van den Brink and Benschop, 2014*).

We do not dismiss the value of good mentoring and recommend that group leaders consider investing time into training and mentoring strategies (see, for example, *Lee et al., 2007*). It was reported recently that ECRs who co-author publications with highly-cited scientists have greater probability of repeatedly co-authoring additional publications with top-cited scientists, and, ultimately, a higher probability of becoming top-cited scientist themselves (*Li et al., 2019*). While this does not directly constitute mentorship, it does provide an indication of the value of being able to follow or mimic an established research leader.

We consider the most concerning of all of our results to be the high rate at which ECRs (41.4% of females and 30.7% of males) claimed that questionable research practices within their institutions had negatively impacted their careers. We did not define 'questionable research practices' in our survey, but this terminology is commonly used to describe activities ranging from fraud to less egregious practices, such as data exclusion or p-Hacking (*John et al., 2012*). A 2019 survey conducted by the National Health

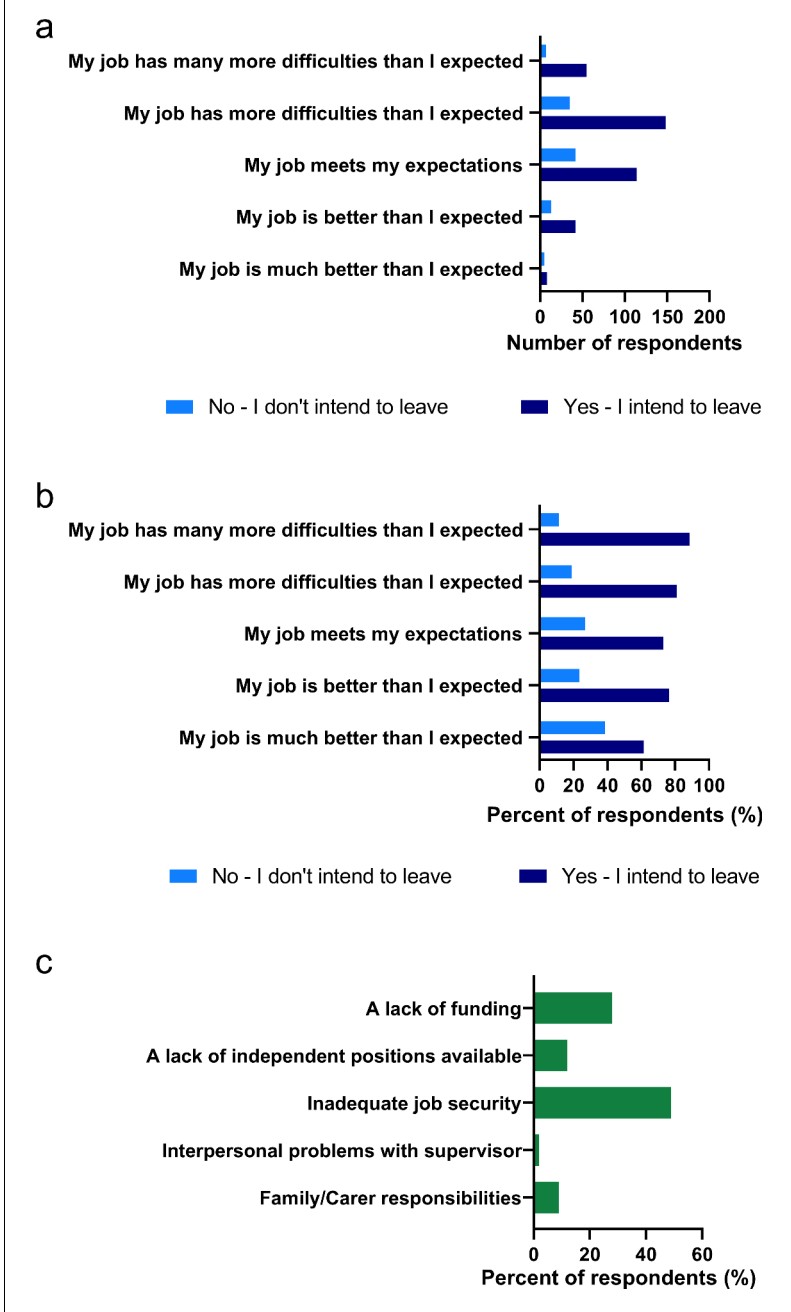

**Figure 5.** ECRs expectations of their current position and their intention to leave. Answer to survey Question 73, '*How does your job as an early-career researcher meet your original expectations?'* (n = 469), and respondents' intention to leave or remain in their position. (**a**) Data shown as raw number of respondents. (**b**) Data shown as percentage of each group of respondents. Note correlation between job expectation and intention to leave (n = 469, regression analysis, p=0.0234). (**c**) These data outline likely reasons for why ECRs would consider leaving a career in research (Question 67 in survey, n = 425, note that 38 answered other and are not accounted for in this graph).

and Medical Research Council (NHMRC) of Australia found that 54% of all survey participants were aware of researchers feeling tempted or under pressure to compromise on research quality, and that junior researchers were most likely to be aware of such instances, while ethics committee members were least likely (*NHMRC Australia, 2020*). Given the very high stress on individual ECRs and on the system, it is rational to expect that rates of questionable research practice could be on the rise. In 2005, Ioannidis reasoned that '*most published research findings are false'*, discussing the influence of data selection bias and financial pressures on data interpretation and reported outcomes (*Ioannidis, 2005*).

Global research pressures have not declined since 2005, and in 2016 Nature published the results from a survey of 1576 researchers on the so-called reproducibility crisis (*Baker, 2016*). This survey found that pressure to publish and selective reporting were perceived to contribute to greater than 60% of reproducibility problems. Our interpretation of the survey data we collected, where ~ 35% of respondents indicated that questionable research practices had impacted their careers, is that the full extent of known misconduct or data reproducibility problems is likely underestimated. Given that ECRs are both sufficiently trained to identify problems, and often in the laboratory enough to observe these problems, concern from this cohort should be viewed as genuine. Future surveys should ask respondents to characterise what types of questionable research practices they believe are most prevalent, and which are the most harmful.

Our findings highlight the need for institutional and national consideration regarding how pressures are playing out in the Australian STEMM research eco-system. We do not blame institutions or individual ECR mentors for these problems. Few ECRs (1.4%) indicated that they would leave their current position because of poor interpersonal relationships with their supervisor. Rather, we consider that the challenges experienced by ECRs in Australia reflect systemic problems. Most ECRs (78.3%) had considered a major career change in the past five years, including leaving academia all together or moving overseas. If ECRs left their current positions it would be primarily because of lack of funding and job security (total 77.1%). When the ECR responses were delineated based on years post PhD, those who were greater than 4 years post PhD were less satisfied than those who were 4 years or less post PhD. Our observations parallel a previous study that observed that job satisfaction was greater for those who had more recently started their first postdoctoral appointment (*Miller and Feldman, 2015*).

In our survey, female ECRs indicated experiencing higher rates of inequitable hiring practices and harassment from those in a position of power than their male counterparts. In contrast, more males felt unsafe in their work environment. We also found that both male and female ECRs were concerned about parental/carer responsibilities, knowing that delayed research productivity could compromise their career prospects. Men were more concerned about this than women, possibly reflecting recent efforts to accommodate mothers, but not necessarily families.

Challenges for researchers are not isolated to Australia. A survey by the Wellcome Trust in 2020 of over 4,000 researchers (mostly in the UK) paralleled many of our observations (*Wellcome Trust, 2020*; *Abbott, 2020*). While 84% of researchers were proud to work in the research community, only 29% felt secure in pursuing a research career, and 23% of junior researchers and students suggested that they had felt pressured by their supervisor to produce a particular result. In agreement with our findings on 'questionable research practices', 43% of respondents in the Wellcome survey believed that their workplace puts more value on meeting research metrics than the quality of the research. It is clear that these are global challenges that will require intervention at all levels of the research community.

### Compounding inefficiencies and suggestions for change

Many current problems in the field could be viewed as inefficiencies. Solving these problems may contribute to improvements at both the personal and community level, thus justifying investment into solutions. For example, a study published in 2015 estimated that $28bn per year is 'spent on preclinical research that is not reproducible – in the United States alone' (*Freedman et al., 2015*). Although we have not seen similar estimates for Australia, findings from the recent NHMRC survey suggests there are reasons to be concerned (*NHMRC Australia, 2020*).

A second major source of inefficiency is the low grant funding success rates, coupled with evidence that current mechanisms for ranking applications is unreliable (*Graves et al., 2011*; *Pier et al., 2018*; *Forscher et al., 2019*). In 2012, researchers spent an average 34 days preparing NHMRC grant proposals, only 21% of which were successful; this means that out of the 550 years invested into all applications (3,272 applications), 435 years were spent on unfunded applications (*Herbert et al., 2013*). These problems are not isolated to the NHMRC; the time-cost of preparing and reviewing grants, coupled with poor ranking reliability, have motivated many to propose transition to outright lotteries (*Adam, 2019*).

Poor funding rates, and the inherent risk that an individual's salary might not be funded for the next calendar year drive high attrition rates. While the constant flow of eager new PhD graduates into the workforce offers a mechanism to replace those who have exited the system, high turnover should be viewed as another potential source of community level inefficiency. While the less expensive labour of PhD students may help

**Table 8.** Quotes from ECRs in the survey explaining why they do not leave academia, and their fears regarding employment outside of the academic workplace.

| Quote number | Specific response |
| --- | --- |
| 1 | *Because it took me so long to earn my PhD, not using it now would seem like a waste. Also, I don't know what else I am qualified to do.* |
| 2 | *I didn't know what the other options were or how to pursue them.* |
| 3 | *I enjoy science. I feel like leaving would be a failure. I try to continue/stay alive until that failure happens.* |
| 4 | *I've spent 10 years training to be an academic. I want to be an academic, but it seems it just isn't my choice at the end of the day. I'll stay until I am no longer competitive. I am keeping my eyes open and looking at other opportunities but so far no one wants me outside academia either.* |
| 5 | *I have no skills in anything else.* |
| 6 | *After 13 years at university, a divorce, my body and mind falling apart, and pulling myself up from grinding childhood poverty and abuse there isn't anything else I feel that I am qualified to do. I am really good at my job yet overqualified and not healthy enough to do anything else. I am stuck here.* |
| 7 | *I also cannot imagine working in another environment, I actually don't know what other options are available and whether these would be fulfilling.* |
| 8 | *I constantly think about leaving academia/research (from necessity not choice) but don't know how and am not qualified for any other jobs.* |

to balance the budgets of individual laboratories, the process of training many individuals for brief careers in science represents an inefficiency likely to negatively impact national research budgets and output. This workforce inefficiency is almost certainly linked to inefficiencies associated with irreproducible science.

Lastly, our data showed that job security (52%) was the number one reason that Australian STEMM ECRs said they might leave their current position, in agreement with the Wellcome survey which also identified this as a major concern. As a community we need to work to improve job security (take care of our people) and the quality of research data (our product). Below we set out a list of national and international recommendations that could help tackle some of these problems.

### Recommendations for Australia

With the goal of stabilising the careers of early-career researchers in science, technology, engineering, mathematics and medicine and maintaining research quality, Australia should consider: (1) An increase in GDP expenditure on research and development to align with other Organization for Economic Co-operation and Development (OECD) nations. (2) Trim PhD completion numbers to better align with current workforce demands. While PhD students offer a sizable and inexpensive workforce, a long-term view of graduate contributions are likely to benefit the field. (3) Research funds should be distributed through smaller and more consistent grants with the goal of supporting the long-term career development of ECRs. Innovation and innovators are rare, and time is required to test ideas and develop gifted researchers. (4) Finally, Australia should establish an independent research ombudsman to oversee research integrity issues. Need for an independent research ombudsman has been discussed previously (*Vaux, 2013*; *Brooks et al., 2016*).

### Institutional recommendations

At the institutional level, around the world, the research environment for early-career researchers in STEMM disciplines could be improved by: (1) Training mentors to manage the career development of ECRs. (2) Aiming to provide greater career stability through longer contracts. (3) Developing skills training programs that prepare PhD candidates and early-career scientists for employment outside of academia for when long-term academic employment is not viable, and a culture for attending this training. (4)

Supporting the development of a research culture that counters questionable research practices by encouraging all academics to ask questions, challenge hype and report honestly.

## Limitations of the study

Our survey captured the opinion of 658 early-career researchers working in Australia in STEMM disciplines. It has proven difficult to determine the precise number of such ECRs. There were approximately 23,000 higher education staff in Level A and B positions (all disciplines) in 2019 (*Department of Education, Skills and Employment, 2019*), but it is not known how many of these were in the STEMM disciplines. Previous work estimated the number of postdoctoral researchers employed in Australia as 6,000 (*Hardy et al., 2016*). It is likely our survey captured opinion from 5–10% of the target population. As the survey participants were self-selected, it is possible that we attracted a disproportionate number of dissatisfied respondents. Surveys were distributed by third parties at research institutes, or recruited via social media, potentially limiting or biasing distribution and preventing calculation of response rate. This process and its limitations have been reported briefly in *Research Ethics Monthly* (*Christian et al., 2019*).

In our survey, we collected some demographic data which could be used to measure diversity. These data included country of birth, language spoken at home, country of PhD, whether respondents lived with a partner (no, heterosexual, same sex or prefer not to say) and chronic disability. However, only 20 (3%) respondents indicated they were in a living with a partner of the same sex, and 3% preferred not to say. Furthermore, we did not collect data on the ethnicity of respondents and therefore cannot know how this may have influenced the findings of our survey. Further research should examine how the challenges identified in this study may vary between diverse groups. We did not observe indications of cultural bias, but this could be because our survey was conducted in English, perhaps selecting for those whose language skill made them well equipped to complete the survey.

## Materials and methods

This survey formed one part of a mixed methods research project which explored challenges faced by early-career scientists at universities and at independent research institutes in

Australia. The primary research questions from which the survey questions were derived were; (1) What are the relationships between ECR job satisfaction or dissatisfaction and their likelihood of staying in STEMM? (2) What are the principal factors that shape the ECR experience of various cohorts in the STEMM in Australia? (3) What are the motivations for ECRs leaving their research position? (4) What are the specific features of the experiences and environment of those ECRs who remain in STEMM? The definition of 'early-career researcher' for the purpose of this project included holding a PhD or equivalent, awarded no more than ten years prior and employment in an Australian university or independent research institute in a STEMM discipline.

### Survey

Survey questions are included in *Supplementary file 1*. Quantitative data was collected from 658 respondents in an on-line survey of ECRs working in a scientific environment in universities and research institutes across Australia. Individuals employed in private enterprise/industry, not-for profit entities or in government funded organisations were excluded from this project as their research environments are considered different. The conceptual framework for the study was built on frameworks for job satisfaction for academics developed by *Rosser, 2004* and *Basak and Govender, 2015*, which identified important factors as workload, job security, job satisfaction, challenges, mentoring and supervision, career planning, intention to leave, career breaks and expectations about the career. Survey questions were selected to explore these factors and were supplemented with questions seeking demographic information which included the institution type, research discipline, country of origin, family situation and work arrangements. In addition, we held focus group discussions which enabled us to identify other important questions, and to optimise our approach. The questionnaire for the survey was developed by first compiling questions, often used in a broader or international context, from research literature including questions from Australian Council of Education Research, The EMCR Forum at the Australian Academy of Science, Federation of Australian Scientific and Technological Societies (FASTS), Global Young Academy, National Science Foundation, Nature Research and Vitae (*Christopherson et al., 2014*; *Hardy et al., 2016*; *Coates et al., 2008*; *Coussens et al., 2017*; *Nature Research*

*and Penny, 2017*; *Bell and Yates, 2015*; *Phou, 2015*; *Vitae, 2018*).

In order to cover all the themes identified in the literature as matters relating to job satisfaction or dissatisfaction. Some additional questions were created if no suitable question was identified elsewhere. Questions were combined and modified to create a question bank for this survey relevant to the research questions and the Australian context and further informed by data collected from a focus group of ECRs, after which the survey was pilot tested. In keeping with the conceptual framework for the study, matters investigated include inequity, bias or discrimination with respect to age, gender, inequitable hiring practices and harassment based on different power positions, mentoring and supervision, career planning, training and professional development and work life balance. The data from these questions were supplemented by questions seeking demographic information which included the institution type, research discipline, country of origin, family situation and work arrangements.

The invitation to take part in the survey was distributed via email after direct contact with the institutions, via social media or 'umbrella groups' such as EMCR Forum (*Australian Academy of Science, 2020*) and The Australian Society for Medical Research (*ASMR, 2020*) with members or affiliates drawn from the STEMM community who were likely to include the target group.

A focus group discussion attended by seven ECRs on January 30, 2019 evaluated the questionnaire prior to the survey and participants in the focus group offered additional insights. These seven focus group participants were ECRs from five STEMM disciplines and four institutions based in Sydney, Australia who responded to an email invitation that was circulated within Sydney institutions and who were considered to be broadly representative of ECRs in STEMM. All provided informed consent. Once the survey was established, a pilot study (n = 22) permitted testing for understanding and clarity and to check for technical difficulties. The pilot survey ran from February 14 to February 28, 2019. The National survey followed, and the data from the survey is discussed in this paper. The survey ran from March 5 to June 14, 2019. The survey was conducted online using LimeSurvey (v2.01). Eligibility to participate was determined by the initial questions in the survey.

## Acknowledgements

Katherine Christian is supported by an Australian Government Research Training Program (RTP) Fee-Offset Scholarship through Federation University Australia. MRD is supported by a NHMRC Fellowship (APP1130013). The Translational Research Institute is supported by Therapeutic Innovation Australia (TIA). The Australian Government supports TIA through the National Collaborative Research Infrastructure Strategy (NCRIS) program. The Authors would like to thank Dr. Kathryn Futrega for critical discussion and figure design.

**Katherine Christian** is in the School of Arts, Federation University Australia, Ballarat, Australia
katherinechristian@students.federation.edu.au
https://orcid.org/0000-0001-9690-1417

**Carolyn Johnstone** is in the School of Arts, Federation University Australia, Ballarat, Australia

**Jo-ann Larkins** is in the School of Engineering, Information Technology and Physical Sciences, Federation University Australia, Churchill, Australia

**Wendy Wright** is in the School of Science, Psychology and Sport, Federation University Australia, Churchill, Australia
https://orcid.org/0000-0003-3388-1273

**Michael R Doran** is in the School of Biomedical Sciences and Centre for Biomedical Technologies within the Queensland University of Technology at the Translational Research Institute, Brisbane, Australia. He is also affiliated with the Mater Research Institute in Brisbane, Australia and the National Institute of Dental and Craniofacial Research at the National Institutes of Health in Bethesda, United States
michael.doran@qut.edu.au
https://orcid.org/0000-0001-5876-4757

**Author contributions:** Katherine Christian, Conceptualization, Data curation, Formal analysis, Validation, Investigation, Methodology, Writing - original draft, Project administration, Writing - review and editing; Carolyn Johnstone, Jo-ann Larkins, Wendy Wright, Conceptualization, Resources, Formal analysis, Supervision, Validation, Methodology, Writing - original draft, Writing - review and editing; Michael R Doran, Conceptualization, Formal analysis, Validation, Methodology, Writing - original draft, Writing - review and editing

**Competing interests:** The authors declare that no competing interests exist.

**Ethics:** Human subjects: This study has been conducted according to the guidelines of the ethical review process of Federation University Australia and the National Statement on Ethical Conduct in Human Research (Approval Number 18/139A). Voluntary and informed participants contributed to these data. All provided informed consent.

## Funding

| Funder | Grant reference number | Author |
|---|---|---|
| National Health and Medical Research Council | APP1130013 | Michael R Doran |

The funders had no role in study design, data collection and interpretation, or the decision to submit the work for publication.

**Decision letter and Author response**
Decision letter https://doi.org/10.7554/eLife.60613.sa1
Author response https://doi.org/10.7554/eLife.60613.sa2

# Additional files

## Supplementary files

- Supplementary file 1. Questions from online survey.
- Supplementary file 2. Comparison of satisfaction data from this survey and historical surveys in Australia.
- Transparent reporting form

## Data availability

The data set for this paper has been uploaded to Federation University Figshare, which is an open access data base. DOI 10.25955/5f98c272a6ef5. These data has been purged of institution name, country of birth and identifying statements made in open text responses. Most of the remaining survey data is likewise be available. Some project data is subject to embargo to protect the anonymity of participants however it may be shared subject to the approval of the Federation University of Australia Human Research Ethics Committee. See https://federation.figshare.com/projects/Challenges_Faced_by_Early-Career_Researchers_in_the_Sciences_in_Australia_and_the_Consequent_Effect_of_those_Challenges_on_their_Careers_a_Mixed_Methods_Project/90317.

The following dataset was generated:

| Author(s) | Year | Dataset URL | Database and Identifier |
|---|---|---|---|
| Christian K | 2020 | https://doi.org/10.25955/5f98c272a6ef5 | Figshare, 10.25955/5f98c272a6ef5 |

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
