## [Decision Letter]

Thank you for submitting your article "Survey of Australian STEMM Early Career Researchers raises concerns about research culture" to *eLife* for consideration as a Feature Article. Your article has been reviewed by three peer reviewers, and the evaluation has been overseen by two members of the *eLife* Features team (Julia Deathridge and Peter Rodgers) The following individuals involved in review of your submission have agreed to reveal their identity: Rebeccah S Lijek (Reviewer #2); Harry Rolf (Reviewer #3); Tanvir Hussain (Reviewer #4).

The reviewers and editors have discussed the reviews and we have drafted this decision letter to help you prepare a revised submission. We hope you will be able to submit the revised version within two months. Please also note the following

i) if your article is accepted, we would like to publish some of the relevant code with the article: please let me know if this will be a problem.

ii) *eLife* does not permit figures in supplementary materials: instead we allow primary figures to have figure supplements. I can suggest how to accommodate the figures in your supplementary materials if your article is accepted.

Summary

In this article, the authors surveyed 658 Australian early-career researchers (ECRs) about their job satisfaction, mentorship, working environment, and motivations to stay or leave science. The results provide a comprehensive picture of the culture of research in Australia and make an important contribution to ongoing discussions around the challenges faced by ECRs.

Major comments to address

1) It would be useful to report relevant data on ethnicity, disability and sexual orientation, if available. Given that a lack of DEI efforts was a previously documented source of job dissatisfaction (Zimmerman, Carter-Sowell and Xu, 2016; Professionals Australia, 2014; Potvin et al., 2018; Else, 2019 and Gewin, 2018), any/all survey data on DEI should be included rather than saying data not shown. It is concerning that 1 in 5 ECRs (17.8%) are dissatisfied with their workplace's approach to inclusion of diverse populations, and this finding is underplayed in the current manuscript, especially without any racial/ethnic data on survey respondents. Is it possible that the 17.8% of respondents who are dissatisfied with DEI are the scientists of colour, and the remaining survey respondents are the white scientists who are privileged to be "unconcerned" with issues of DEI in the workplace? That discrimination is a minority problem is exactly the problem. Especially since Table 3A/3B shows that ~30-40% of respondents have suffered harassment, it seems important to break this down by other demographic info (like race/ethnicity, age, parental status etc.). Discrimination and harassment is a serious problem that deserves further analysis and discussion in the manuscript.

If the data on the respondents' ethnicity and sexual orientation is not available, this should be discussed as a limitation of the study. Furthermore, it would be interesting to include the data and analysis from Qs 36 in the survey, asking how people's attitudes towards their gender, sexual orientation, or ethnicity impacted their career advancement.

2) The rationale for selecting the topics investigated in the survey is unclear. Please provide justification for the topics selected for the survey and why other relevant pressures on STEM ECRs were omitted (e.g. racism in STEM and demographic data on race/ethnicity). Additionally, please add an explanation of what is new about this study compared to previous surveys . Specifically, describe the findings of the reference cited in the Introduction, from which the survey questions used in your study were sourced.

3) Please make the following changes to the “Suggestions for Change” section:

– The current recommendations provided, although valid, are very high-level and do not provide enough guidance for senior managers in the sector who want to implement changes. It would be great to have more practical recommendations similar to the ombudsman recommendation. For example, the recommendation "aim to provide greater career stability", would benefit from having more information on how institutes should do this, such as longer contracts or ideas on how to make performance reviews more effective.

– The recommendations suggest several systematic changes at an international, national and institutional level. While changes like increased funding, regulation, a greater focus on research quality/accountability or reducing the number of PhDs may improve the research system, the section does not clearly explain how adopting these recommendations will improve the experience of ECRs or how the survey results led to these being necessary/desirable changes. For example, in Australia PhDs account for 57% of the R&D human resources dedicated to research and development, reducing the number would have significant unintended consequences to the research system the consideration of which goes beyond the scope of the article (https://www.abs.gov.au/ausstats/abs@.nsf/mf/8111.0). Making such broad and poorly considered recommendations will detract from the quality of the article and its core message.

– This section does not mention who is responsible for implementing recommendations at the different levels. The recommendations could instead pay attention to research culture and associated issues identified by the survey results and consider the challenges and opportunities available at different levels to address them. i.e. what are the benefits of a national or international mentoring scheme as opposed to an institutional one, why is one approach preferred and where does responsibility sit? This would produce recommendations useful to policy makers interested in pursuing the suggestions for change.

4) Please provide a much clearer definition of participant inclusion and exclusion criteria, especially with regard to job categories. Sometimes the authors equate this study population with postdocs but I was under the impression that their definition of ECR as 0-10yrs post PhD could also include independent investigators (like PIs, faculty, or non-trainee staff scientists). The text seems to suggest some respondents are running their own laboratories and are no longer trainees. I would appreciate some clarity on the Australian system and whether faculty/PIs were excluded or included in this sample. Much of the data interpretation relies on knowing what jobs respondents have and so analyses should include a stratification of survey responses by job (e.g. trainee/postdoc vs. independent PI/staff scientist).

5) In regards to the word cloud in Figure 1, please use the “thematic analysis” approach as described here [https://www.scribbr.com/methodology/thematic-analysis/], and an appropriate quantitative text analysis approach to support/explain the themes. One possible approach for this quantitative analysis would be to apply the algorithm used to generate the word cloud to visualise adjectives/verbs in the text responses or other approaches such as word co-occurrence – see the open source software KH Coder for examples https://khcoder.net. Please avoid the inclusion of words which are not relevant to the data you are trying to extract from the question such as “research”, “want” or “Australia”. The methodology used to analyze the text response to question 76, should also be included in the text.

6) The authors state that "publications [should] outline the limitations of their studies" yet there is no discussion of their own study's limitations. Please include a section outlining the limitations of this study. This section should mention (in addition to other limitations):

– Without any comparator groups for non-ECRs, it is difficult to say that any of these analyses are specific to the ECR population vs. a general phenomenon common to the entire STEM workforce.

– The article would benefit from some reflection by the authors on the limitations of the survey method (noting a statistically significant sample was not achieved) and the interpretive analysis/description by the researchers (i.e. no follow-up with participants is mentioned, why were results not explored in more depth / were there plans to do so?). It could also discuss any steps taken to address limitations along with lessons learnt.

7) The data on ECRs' considerations of job change (Figure 2B,C) does not seem to separate out attrition/stagnation (leaving the job because a contract runs out or lateral moves to another training position) vs. promotion into more senior roles like π (the stated desired outcome of said PhD/postdoctoral training). Doesn't the survey response "Yes, to take another position in the same field of science in Aus./overseas" include both negative and positive career outcomes? As a result, I find it challenging to interpret these data as "job insecurity" and question why the authors only interpret this as a negative. The same critique can be applied to the data that career moves are common without disambiguation of moves due to career stagnation vs. career advancement

8) I share the authors' concern about the reproducibility crisis in science and the high rates of "questionable research practices" reported in this study. Yet I find it hard to connect their data to that crisis because of the lack of definition of the term "questionable research practices" in the survey. The authors rightly mention this lack of definition. Respondents seem to think that a "questionable research practice that negatively impacted my career" includes the (abhorrent) bullying described in Table 4 Quote 4 – but the fact that the authors confound this with the data fraud (e.g. falsification or exclusion) that leads to the reproducibility crisis is overinterpretation. I don't doubt that data fabrication and exclusion occurs, and is a major problem in STEMM; I just am not convinced this survey question captures it with enough precision to interpret these data that way. I think it's important to provide evidence of research misconduct accurately and specifically, because otherwise it can lead to fear-mongering, anti-science sentiment that disproportionately undermines trust in the scientific endeavour.

Please can you remove the section in the discussion linking together the result from the survey on questionable research practices and the reproducibility crisis, or add a few sentences explaining how the lack of definition of the term “questionable research practices” makes this link difficult to determine directly, and what further work would be needed to research this finding further.

9) The Introduction focuses on the issue of employability and research quality as drivers for the work. While these two issues do emerge in the survey results, and are good reasons to be concerned about the position of ECRs in the research workforce, the section would benefit from a broader discussion of research culture to situate the findings of the survey. Discussion about perceptions of research quality could consider results from a related survey by the NHMRC the NHMRC 2019 Survey of research culture in Australian NHMRC-funded institutions https://www.nhmrc.gov.au/research-policy/research-quality

10) Please can you add a methods and procedure section before the results. This should include the section on how Survey was conducted and respondents were recruited which can be found in the Materials and methods. It would also be good to include how the questions for the survey were chosen and the steps taken to ensure there were no biases in the questionnaire.

Please can you also include the first section of the results, “The respondent demographics”. The Results section would then start with, “Characteristics that influence ECR job satisfaction”.

11) Where are the data described in subsection “Impact of Inappropriate behaviours”? I think this stratification is very interesting and may demonstrate burn-out and accumulated years of being undervalued in the workforce. If this data is not already shown, please can you include it as a figure in the manuscript.

---

## [Author Response]

Major comments to address1) It would be useful to report relevant data on ethnicity, disability and sexual orientation, if available. Given that a lack of DEI efforts was a previously documented source of job dissatisfaction (Zimmerman, Carter-Sowell and Xu, 2016; Professionals Australia, 2014; Potvin et al., 2018; Else, 2019 and Gewin, 2018), any/all survey data on DEI should be included rather than saying data not shown. It is concerning that 1 in 5 ECRs (17.8%) are dissatisfied with their workplace's approach to inclusion of diverse populations, and this finding is underplayed in the current manuscript, especially without any racial/ethnic data on survey respondents. Is it possible that the 17.8% of respondents who are dissatisfied with DEI are the scientists of colour, and the remaining survey respondents are the white scientists who are privileged to be "unconcerned" with issues of DEI in the workplace? That discrimination is a minority problem is exactly the problem. Especially since Table 3A/3B shows that ~30-40% of respondents have suffered harassment, it seems important to break this down by other demographic info (like race/ethnicity, age, parental status etc.). Discrimination and harassment is a serious problem that deserves further analysis and discussion in the manuscript.If the data on the respondents' ethnicity and sexual orientation is not available, this should be discussed as a limitation of the study. Furthermore, it would be interesting to include the data and analysis from Qs 36 in the survey, asking how people's attitudes towards their gender, sexual orientation, or ethnicity impacted their career advancement.

We collected some demographic data which could be used to measure diversity. These data included country of birth, language spoken at home, country of PhD, whether respondents lived with a partner (no, heterosexual, same sex or prefer not to say) and chronic disability. We found little difference between responses about challenges in the workplace from respondents who are Australian born compared with people from elsewhere. In addition to the data already contained in Figure 3—figure supplement 1C, we have added data in Table 2.

We did not directly ask about sexual orientation other than in Q11 *Do you live with a partner or spouse?* where an option was “yes with a same sex partner”. This option was included at the suggestion of Science in Australia Gender Equality (SAGE), with the aim of capturing whether parents in a same sex relationship had different or additional issues with respect to childcare or parental leave when compared with heterosexual parents. Only 20 (3%) respondents indicated they were in a live-in same sex relationship and there were no comments made which provided further information for people from this sector.

We did not ask respondents about disability, however Q72 *Do you have a long term health condition or disability that restricts you in your everyday activities and has lasted, or is likely to last, for more than 6 months?* could have captured this data. We did learn from comments that it was difficult for people with a chronic health condition to relocate.

While an examination of the difficulties of diverse populations is certainly merited for further research, we did not investigate experiences for this sector any further in this project.

We have now included a comment regarding the impact of chronic health conditions.

“Within the small number of respondents that reported a chronic health condition (12%) some indicated that relocation was challenging.”

2) The rationale for selecting the topics investigated in the survey is unclear. Please provide justification for the topics selected for the survey and why other relevant pressures on STEM ECRs were omitted (e.g. racism in STEM and demographic data on race/ethnicity). Additionally, please add an explanation of what is new about this study compared to previous surveys. Specifically, describe the findings of the reference cited in the Introduction, from which the survey questions used in your study were sourced.

The conceptual framework for the study was built on frameworks for job satisfaction for academics developed by Rosser, 2004 and Basak and Govender, 2015 which identified important factors as workload, job security, job satisfaction, challenges, mentoring and supervision, career planning, intention to leave, career breaks and expectations about the career. These questions were supplemented with questions seeking demographic information which included the institution type, research discipline, country of origin, family situation and work arrangements. In addition, we held focus group discussions which enabled us to identify other important questions, and to optimise our approach.

The following text has been added to the Materials and methods to clarify these details:

“The conceptual framework for the study was built on frameworks for job satisfaction for academics developed by Rosser, 2004 and Basak and Govender, 2015 which identified important factors as workload, job security, job satisfaction, challenges, mentoring and supervision, career planning, intention to leave, career breaks and expectations about the career. Survey questions were selected to explore these factors and were supplemented with questions seeking demographic information which included the institution type, research discipline, country of origin, family situation and work arrangements. In addition, we held focus group discussions which enabled us to identify other important questions, and to optimise our approach.”

3) Please make the following changes to the “Suggestions for Change” section:– The current recommendations provided, although valid, are very high-level and do not provide enough guidance for senior managers in the sector who want to implement changes. It would be great to have more practical recommendations similar to the ombudsman recommendation. For example, the recommendation "aim to provide greater career stability", would benefit from having more information on how institutes should do this, such as longer contracts or ideas on how to make performance reviews more effective.– The recommendations suggest several systematic changes at an international, national and institutional level. While changes like increased funding, regulation, a greater focus on research quality/accountability or reducing the number of PhDs may improve the research system, the section does not clearly explain how adopting these recommendations will improve the experience of ECRs or how the survey results led to these being necessary/desirable changes. For example, in Australia PhDs account for 57% of the R&D human resources dedicated to research and development, reducing the number would have significant unintended consequences to the research system the consideration of which goes beyond the scope of the article (https://www.abs.gov.au/ausstats/abs@.nsf/mf/8111.0). Making such broad and poorly considered recommendations will detract from the quality of the article and its core message.– This section does not mention who is responsible for implementing recommendations at the different levels. The recommendations could instead pay attention to research culture and associated issues identified by the survey results and consider the challenges and opportunities available at different levels to address them. i.e. what are the benefits of a national or international mentoring scheme as opposed to an institutional one, why is one approach preferred and where does responsibility sit? This would produce recommendations useful to policy makers interested in pursuing the suggestions for change.

We are grateful for the opportunity to provide further comment on how systems can be improved. While we cannot dictate policy, we feel that we can add value by making the inefficiencies in the current research system at the personal and community level clear. With this objective, we have added text to subsections “Compounding inefficiencies and suggestions for change”, “Recommendations for Australia” and “Institutional recommendations”

4) Please provide a much clearer definition of participant inclusion and exclusion criteria, especially with regard to job categories. Sometimes the authors equate this study population with postdocs but I was under the impression that their definition of ECR as 0-10yrs post PhD could also include independent investigators (like PIs, faculty, or non-trainee staff scientists). The text seems to suggest some respondents are running their own laboratories and are no longer trainees. I would appreciate some clarity on the Australian system and whether faculty/PIs were excluded or included in this sample. Much of the data interpretation relies on knowing what jobs respondents have and so analyses should include a stratification of survey responses by job (e.g. trainee/postdoc vs. independent PI/staff scientist).

ECRs were defined as being less than 10 years since PhD completion, similar to the definition used by the Global Young Academy in their study of how to best support young scientists on a global scale. This information was provided in the Introduction, and we have added this information to the Materials and methods section as well. It is correct that the eligible researchers can fall into the range of raw trainee to independent investigator: years postdoctoral is the prime definition for inclusion. As shown in the demographic data their positions include research only, research and teaching, teaching only and clinician researcher.

The following text was added to the Materials and methods:

“We defined ECRs as being less than 10 years since PhD completion, similar to the definition used by the Global Young Academy in their study of how to best support

young scientists on a global scale (Pain, 2014; Bell et al., 2015).”

5) In regards to the word cloud in Figure 1, please use the “thematic analysis” approach as described here [https://www.scribbr.com/methodology/thematic-analysis/], and an appropriate quantitative text analysis approach to support/explain the themes. One possible approach for this quantitative analysis would be to apply the algorithm used to generate the word cloud to visualise adjectives/verbs in the text responses or other approaches such as word co-occurrence – see the open source software KH Coder for examples https://khcoder.net. Please avoid the inclusion of words which are not relevant to the data you are trying to extract from the question such as “research”, “want” or “Australia”. The methodology used to analyze the text response to question 76, should also be included in the text.

The word cloud was generated using NVivo v12 for Mac, an accepted analysis tool which helps researchers analyse data. The query facility in NVivo 12 for Mac which permits a count of word frequencies and generation of word cloud images was used to provide illustrations of the themes which were emerging from open text responses for selected questions in the on-line survey and from a section of the focus group discussion. Images were generated for the top 500 words with a minimum length of three letters long and including stemmed words. In addition to the standard “stop words”, the names of interview participants were excluded. Because this is not a sophisticated portion of the analysis, we have simply cited the software in the figure caption, and made clear in the results from which question the data was harvested.

While we appreciate the suggestion to remove words such as “research” or “Australia”, we feel that these words are specifically important in the context of our respondents. In an effort to meet the suggestions of the reviewers we have now introduced Table 2 in addition to the word cloud, where the words are limited to only those words which indicate satisfaction or dissatisfaction.

6) The authors state that "publications [should] outline the limitations of their studies" yet there is no discussion of their own study's limitations. Please include a section outlining the limitations of this study. This section should mention (in addition to other limitations):

Thank you for the opportunity to discuss the study limitations, and suggestions for what future studies might consider.

The following text has been added to the Discussion.

**“**Limitations of the study

Our survey captured the opinion of 658 early-career researchers working in Australia on STEMM subjects. […] This process and its limitations have been reported briefly in Research Ethics Monthly (Christian et al., 2019).”

– Without any comparator groups for non-ECRs, it is difficult to say that any of these analyses are specific to the ECR population vs. a general phenomenon common to the entire STEM workforce.

To provide context for our survey observations, we have included a summary table that describes outcomes from other Australian surveys, and the following text in the Results section.

“We compared our survey respondents’ satisfaction data with previous survey data from Australian academics (Supplementary file 2; Coates et al., 2009; Bexley et al., 2011; Bell et al., 2015; National Tertiary Education Union, 2017). Each of these studies used one or more of the “job satisfaction” questions from our survey in their survey of the Australian academic workforce. It can be seen respondents from the current study are more concerned about job security than respondents in any of the other studies. Our respondents also indicated a higher level of personal stress (52%) than those in all the other studies (28%-43%) and agree most strongly (65%) “this is not a good time for any young person to aspire to an academic career”. Their reported job satisfaction is low (62%). The combination of answers to these questions for current ECRs relative to those for the other studies indicates a situation about which there should be grave concern.”

– The article would benefit from some reflection by the authors on the limitations of the survey method (noting a statistically significant sample was not achieved) and the interpretive analysis/description by the researchers (i.e. no follow-up with participants is mentioned, why were results not explored in more depth / were there plans to do so?). It could also discuss any steps taken to address limitations along with lessons learnt.7) The data on ECRs' considerations of job change (Figure 2B,C) does not seem to separate out attrition/stagnation (leaving the job because a contract runs out or lateral moves to another training position) vs. promotion into more senior roles like π (the stated desired outcome of said PhD/postdoctoral training). Doesn't the survey response "Yes, to take another position in the same field of science in Aus./overseas" include both negative and positive career outcomes? As a result, I find it challenging to interpret these data as "job insecurity" and question why the authors only interpret this as a negative. The same critique can be applied to the data that career moves are common without disambiguation of moves due to career stagnation vs. career advancement

We agree it is possible that the outcomes in this section could be either positive or negative for the respondent. In Figure 4C we provide a breakdown of why respondents might leave a research position. The most commonly cited reason was job insecurity. Job security is a problem even for those who are happy and productive in their current position. The reviewer is of course correct that some movement or change of jobs will be motivated by efforts to advance careers, or to capture opportunities to travel with career development. To make this clear, we have added additional text to the section of the manuscript that discusses this issue.

“The need to relocate

Many academics relocate to capture job opportunities, and many appreciate the opportunity to move internationally with their career. However, we observed that the academic culture promotes a perceived *need* to relocate during the ECR years, and that many ECRs who wished to remain in academia considered moving as part of their career development process.”

8) I share the authors' concern about the reproducibility crisis in science and the high rates of "questionable research practices" reported in this study. Yet I find it hard to connect their data to that crisis because of the lack of definition of the term "questionable research practices" in the survey. The authors rightly mention this lack of definition. Respondents seem to think that a "questionable research practice that negatively impacted my career" includes the (abhorrent) bullying described in Table 4 Quote 4 – but the fact that the authors confound this with the data fraud (e.g. falsification or exclusion) that leads to the reproducibility crisis is overinterpretation. I don't doubt that data fabrication and exclusion occurs, and is a major problem in STEMM; I just am not convinced this survey question captures it with enough precision to interpret these data that way. I think it's important to provide evidence of research misconduct accurately and specifically, because otherwise it can lead to fear-mongering, anti-science sentiment that disproportionately undermines trust in the scientific endeavour.Please can you remove the section in the discussion linking together the result from the survey on questionable research practices and the reproducibility crisis, or add a few sentences explaining how the lack of definition of the term “questionable research practices” makes this link difficult to determine directly, and what further work would be needed to research this finding further.

We understand the reviewer’s concerns, and do not want to promote fear-mongering or anti-science sentiment. We have modified the statements made in the conclusion both with caveats, and additional data from another recent Australian survey. At the end of this paragraph we recommend that future surveys ask respondents for further feedback on questionable research practices. We feel that the recent survey data from National Health and Medical Research Council (NHMRC) of Australia does indicate that broad concern in the community about these issues, and we hope that our data and analysis will motivate further conversations about how to manage these problems. While we do not expand on this in the manuscript, it is conceivable that these problems are being amplified by job insecurity driven by the COVID-19 crisis.

“We did not define “questionable research practices” in our survey, but this terminology is commonly used to describe activities ranging from fraud to less egregious practices, such as data exclusion or p-Hacking (John et al., 2012). […] Future surveys should ask respondents to characterise what types of questionable research practices they believe are most prevalent, and which are the most harmful.”

9) The Introduction focuses on the issue of employability and research quality as drivers for the work. While these two issues do emerge in the survey results, and are good reasons to be concerned about the position of ECRs in the research workforce, the section would benefit from a broader discussion of research culture to situate the findings of the survey. Discussion about perceptions of research quality could consider results from a related survey by the NHMRC the NHMRC 2019 Survey of research culture in Australian NHMRC-funded institutions https://www.nhmrc.gov.au/research-policy/research-quality

Thank you for pointing us to these survey results. We have discussed and highlighted some of their observations in the manuscript Discussion. These comments are as described for the question above. Additionally, we have included Appendix 1—table 1 to enable job satisfaction data from our survey to be compared to other National surveys.

10) Please can you add a methods and procedure section before the results. This should include the section on how Survey was conducted and respondents were recruited which can be found in the Materials and methods. It would also be good to include how the questions for the survey were chosen and the steps taken to ensure there were no biases in the questionnaire.

The Materials and methods section has been moved to before the Results as requested. Further, additional text has been added to the Materials and methods to provide clarity.

“The conceptual framework for the study was built on frameworks for job satisfaction for academics developed by Rosser, 2004 and Basak and Govender, 2015 which identified important factors as workload, job security, job satisfaction, challenges, mentoring and supervision, career planning, intention to leave, career breaks and expectations about the career. Survey questions were selected to explore these factors and were supplemented with questions seeking demographic information which included the institution type, research discipline, country of origin, family situation and work arrangements. In addition, we held focus group discussions which enabled us to identify other important questions, and to optimise our approach.”

Please can you also include the first section of the results, “The respondent demographics”. The Results section would then start with, “Characteristics that influence ECR job satisfaction”.

This minor change has been made.

11) Where are the data described in subsection “Impact of Inappropriate behaviours”? I think this stratification is very interesting and may demonstrate burn-out and accumulated years of being undervalued in the workforce. If this data is not already shown, please can you include it as a figure in the manuscript.

We have added an additional table.